# Genome-wide identification and functional characterization of NPR1-like genes in *Actinidia deliciosa*

Weimin Zhong[1]☯, Yuexia Wang[2,3]☯, Shiming Han[2,3]☯, Jihong Dong[3], Yumei Fang[2]*, Xiaoling Xu[2], Muhammad Umar Rasheed[4], Aiman Malik[4], Qurban Ali[4]*, Muhammad Ashfaq[4], Jia Zhou[1]

1 Guizhou Fruit Science Research Institute, Guizhou Academy of Agricultural Sciences, Guiyang, Guizhou, China, 2 School of Biological Sciences and Technology, Liupanshui Normal University, Liupanshui, P.R. China, 3 School of Public Administration, China University of Mining and Technology, Xuzhou, Jiangsu, China, 4 Department of Plant Breeding and Genetics, University of the Punjab, Lahore, Pakistan

☯ Shiming Han, Weimin Zhong and Yuexia Wang contribute equally
* xinxiang324@sohu.com (YF), saim1692@gmail.com (QA)

## Abstract

*NPR1* (Nonexpresser of Pathogenesis-Related Genes 1) is a central regulator of salicylic acid (SA)-mediated defense signaling in plants and plays a pivotal role in modulating systemic acquired resistance (SAR). Despite its functional importance in biotic stress responses, a comprehensive understanding of the *NPR1* gene family in *Actinidia deliciosa* (kiwifruit) has been lacking. In this study, we performed a genome-wide identification and characterization of *NPR1*-like genes in *A. deliciosa*, identifying five candidate genes (*AdNPR1–AdNPR5*) containing conserved BTB/POZ and ankyrin repeat domains of NPR1 Protein. Phylogenetic analysis revealed that *AdNPR3* and *AdNPR4* grouped closely with *AtNPR3/AtNPR4*, indicating possible sub-functionalization related to pathogen-specific defense signaling. Conserved motif and gene structure analyses indicated strong structural conservation, while promoter analysis revealed diverse cis-regulatory elements associated with hormonal and stress responsiveness. Ka/Ks analysis suggested that the gene family evolved under strong purifying selection, with divergence events dating back to ~625 million years ago. Synteny and dual synteny mapping with *Arabidopsis thaliana, Theobroma cacao,* and *Oryza sativa* indicated segmental duplication as the primary driver of gene family expansion. Transcriptome profiling under *B. cinerea* (fungal) and *Pseudomonas syringae pv. actinidiae* (bacterial) infection demonstrated differential expression, particularly the upregulation of *AdNPR3* and *AdNPR4*, indicating their role in pathogen-induced defense responses. Gene Ontology enrichment and protein–protein interaction network analyses further confirmed the involvement of these genes in SA signaling, immune regulation, and floral development. This study provides foundational insights into the structural, evolutionary, and functional characteristics of

**Data availability statement:** All relevant data are within the manuscript and its Supporting Information files.

**Funding:** Project of Liupanshui Normal University (No.LPSSYKYJJ201601; LPSSY2023XKTD09)and the Science and Technology project of Liupanshui City (Grant #52020-2020-0906).

**Competing interests:** The authors have declared that no competing interests exist.

*NPR1*-like genes in *A. deliciosa* and highlights their potential as molecular targets for improving disease resistance in kiwifruit.

## Introduction

The Non-expresser of Pathogenesis- Related genes 1(*NPR1*) is a pivotal regulatory of systemic acquired resistance (SAR) in plants, functioning as a transcriptional co-activator in salicylic acid (SA)-dependent defense signaling [1]. It plays an essential role in activating downstream pathogenesis-Related (PR) genes, such as PR1, which are critical in establishing long-lasting immunity against a wide spectrum of pathogens, particularly biotrophic and hemi-biotrophic fungi and bacteria [2].

Structurally, *NPR1* comprises three major functional domains. The BTB/POZ (Broad complex, Tram track, and Bric-a-brac/Pox virus and Zinc finger) domain located near the N-terminus is vital for *NPR1* oligomerization and subsequent translocation into the nucleus upon SA accumulation [3]. This domain facilitates *NPR1's* proteasome-mediated degradation, enabling tight post- translational regulation. Secondly the ankyrin repeat domain, composed of four conserved repeats, is crucial for protein- protein interactions- especially with TGA transcription factors that directly bind DNA to initiate PR gene transcription. Thirdly, the *NPR1*-like C domain, often underexplored, is suggested to have roles in redox sensing and is thought to interact with other C-terminal repressors or co-factors for specially tuning in defense signaling [4].

The activation of *NPR1* is tightly regulated by the plants cellular redox state. In a non-induced state, *NPR1* exists as a cytosolic oligomer through intermolecular disulfide bonds. Upon pathogenic attack, cellular reduction breaks these bonds, allowing monomeric *NPR1* to translocate to the nucleus [5]. Once inside, *NPR1* interacts with TGA transcription factors, promoting expression PR genes and other defense-related pathways, thereby strengthening host resistance [2]. Its function extends beyond SAR; it is also involved in cross-communication with jasmonic acid (JA) and ethylene (ET) signaling pathways, often modulating the antagonism between SA and JA response [1,2,6].

The broad conservation and functional redundancy among *NPR1*-like gene family members across dicot and monocot plants indicate its evolutionary significance. Functional homologs of *NPR1* in various crops- such as rice (*OsNPR1*) [7], tobacco (*NtNPR1*) [8] cocoa (*TcNPR1*) [2]and kiwifruit (*AcNPR1/AeNPR1*) [9]display similar defense functions, supporting its use in biotechnological approaches to breed disease-resistant plant [9].

*Actinidia deliciosa*, a commercially valuable kiwifruit species, is susceptible to a range of fungal pathogens that compromise yield and postharvest quality. Key fungal diseases include Botrytis cinerea (gray mold), Sclerotinia sclerotiorum, Diaparthe nobilis, and Alternaria spp. Of these, gray mold caused by Botrytis cinerea is one of the most devastating diseases, causing up to 50% pre-harvest yield loss and rapid post-harvest soft-rot that shortens cold-storage life by 30–40%.It typically initiates at senescent flower parts and infected wounds and then invades the fruit, causing soft

rot and decay, especially under cool and humid condition [10]. The infection affects not only the external surface but penetrates vascular tissues, accelerating ripening and spoilage.

Recent transcriptomic studies on Actinidia species reveal that *NPR1* is overregulated during fungal invasion [2], especially under SA-triggered SAR pathways. *NPR1* activation leads to the expression of PR genes and other defense-related enzymes such as peroxidases, chitinases, and β-1,3-glucanases, which directly inhibit fungal growth or strengthen the cell wall against penetration [11]. Moreover, exogenous treatments that mimic SAR-such as salicylic acid, benzothiadiazole (BTH), or calcium chloride- have been shown to upregulate *NPR1* and confer partial resistance to Botrytis and Diaporthe spp. In postharvest Kiwifruit [10,12]. Therefore, targeting the *NPR1* pathway presents a promising strategy for improving both pre- and post-harvest fungal disease management in *A. deliciosa.*

Kiwifruit (*Actinidia deliciosa*) is a climacteric, dioecious fruit crop native to China but extensively cultivated in New Zealand, Italy, Iran, Greece and Chile. Belonging to the family Actinidiaceae, Kiwifruit is known for its high nutritional value, being exceptionally rich in Vitamin C, dietary fiber, potassium, antioxidants, and actinidin- a unique proteolytic enzyme [13]. The fruits' economic significance has grown tremendously due to rising global demand, particularly in health-conscious markets.

According to the latest FAO statistics [14], global kiwifruit production surpassed 4.4 million metric tons, with China accounting for over 53% followed by Italy, New Zealand and Iran. New Zealand remains the top exporter, with over $1.8 billion USD in export value, driven by the commercial success of Zespri-branded cultivars. The fruit is cultivated under diverse agro climatic conditions, primarily in temperate and sub-tropical zones, with increased emphasis on sustainable and disease-resilient production systems.

Despite its economic importance, kiwifruit faces significant challenges from biotic stresses, especially fungal and bacterial pathogens. These adversely affect yield and postharvest quality, necessitating enhances breeding strategies and transcriptomic studies focusing on key regulatory genes, such as *NPR1*, are vital for developing disease-resistant cultivars and ensuring global food security in horticultural systems.

## Materials and methods

### NPR1 Sequence retrieval from database and BLAST-P in *Actinidia deliciosa*

Sequence retrieval and BLAST-P were conducted to identify candidate *NPR1*-like genes in *A. deliciosa* based on domain conservation. The regulatory protein (NPR1) sequence was retrieved from *Arabidopsis thaliana* from the Protein database of National Center for Biotechnology Information (NCBI) (https://www.ncbi.nlm.nih.gov/). The Gene ID of the sequence: 842733, which was updated on 19-Feb-2025 on NCBI database. This retrieved sequence was validated by using motif finder, an online database (https://www.genome.jp/tools/motif/). After validating the peptide sequence, the online kiwi genome database, Kiwifruit PanGenome Database (https://kiwifruitgenome.atcgn.com/) was used for BLAST-P program with default parameters (e-value: 1e-05, matrix: BLOSUM62, gap-open: 11, gap-extend: 1, filter: F). For further confirmation, the peptide sequence of *A. deliciosa* was used in TBtools for BLAST-P program as done by [2]. All peptide sequences from both methods were re-validated by using motif finder to confirm the domains of *NPR1* genes in all sequences

### NCBI-CDD (Conserved Domain Database) analysis

NCBI-CDD domain analysis aimed to confirm the presence of essential functional domains (BTB/POZ, ankyrin repeats, NPR1-like C) required for NPR1 protein activity. NCBI-CDD (https://www.ncbi.nlm.nih.gov/Structure/cdd/wrpsb.cgi) was used to analyze the presence of these 3 domains in *NPR1* gene for its proper functioning against NCBI-CDD search ID: QM3-qcdsearch-B45C7B96EDCFBB5. Retrieved peptide sequences were used with e-value of 0.010000 and result mode to concise. The visualize domain pattern (from NCBI- batch CDD) under bio-Sequence structure illustrator section in TBTool was used to visualize the hitdata file from NCBI-CDD.

## Phylogeny Analysis of *NPR1* gene in *A. deliciosa* with of different species

Phylogenetic analysis was performed to infer evolutionary relationships and potential functional divergence among *AdNPRs* and *NPR1* homologs in other species. For the analysis of phylogeny of *NPR1* gene in *A. deliciosa* with *NPR1* genes of other species, the phylogenetic tree was constructed. An offline software, MEGA11 was used to align the peptide sequences of NPR1 in *A. deliciosa* and other NPR1 peptide sequences of species like Arabidopsis (*A. thaliana*), Wheat (*T. aestivum*), Rice (*O. Sativa*), Peanut (*A. Hypogea*), Sorghum (*S. bicolor*), Cotton (*G. hirsutum*), Maize (*Zea mays*), Canola (*B. rapa*), Soybean (*G. max*), and Papaya (*C. papaya*) by using Align-by-MUSCLE program. The Phylogenetic tree was constructed by using Neighborhood Joining Algorithms in MEGA11 with 1000 bootstrap replications method. Newick file from MEGA11 was further visualized and present the phylogenetic tree by using an online tool, iTOL (https://itol.embl.de/upload.cgi) as done by [2,15].

## Physio-chemical properties and Subcellular Localization of *NPR1* genes in *A. deliciosa*

Physio-chemical and subcellular localization analysis helped predict the stability, solubility, and cellular location of AdNPRs proteins to infer their biological behavior. Physiochemical properties of *NPR1* genes in *A. deliciosa* were found by using an online platform Expasy Protparam (https://web.expasy.org/protparam/). It includes the numbers of amino acids in protein, protein length, molecular weight of protein, GRAVY value of proteins, Isoelectric point of proteins, Instability index, and Aliphatic index of retrieved proteins in plants [16].

To identify the presence and functional Subcellular location of protein in cells, an online database named as WoLF PSORT (https://wolfpsort.hgc.jp/) was used. Details of original Gene IDs, Chromosomal location (Start and End), and Chromosomal length were determined by Kiwifruit PanGenome Database (https://kiwifruitgenome.atcgn.com/).

## Identification and characterization of Gene Structure of *NPR1* in *A. deliciosa*

**Cis-Regulatory Elements (CREs) identification of *NPR1* in *A. deliciosa*.** There are three main regions/structures of a gene known as a Promotor, an Open Reading Frame (ORFs) and a terminator region. Cis-regulatory element (CRE) analysis was performed to identify stress- and hormone-responsive elements in promoter regions, suggesting regulatory control mechanisms. Promotor sequence was retrieved from Kiwifruit PanGenome Database with 1200 upstream nucleotides. For characterization of that promotor regions, PLANT CARE database (https://bioinformatics.psb.ugent.be/webtools/plantcare/html/) was used on 23rd April 2025 at 16:57 Pakistan (UTC + 5). The database was used to identify Cis-Regulatory Elements (CREs) within the promotor region to characterize a gene's function. File from PLANT CARE were further visualized by using Basic Bio-sequence view in TBtool. Further a heatmap was constructed to identify the numbers of a specific CRE in promotor regions of that gene which control the regulation of genes..

**Motif and intron exon analysis *NPR1* in *A. deliciosa*.** Motifs and Introns & Exons are characterized within ORF regions. Motif analysis was used to detect conserved sequence motifs associated with specific functional domains, informing potential defense-related roles. For motif identification, an online database MEME-Suite (http://meme.sdsc.edu/meme/website/intro.html) was used. The peptide sequences of NPR1 proteins were used with max motif finding value at 20 as done in [15]. The file MEME.xml downloaded and visualized by using Graphics section in TBtool. These motifs were also validated on Kiwifruit PanGenome Database.

Intron-exon structure analysis aimed to evaluate gene architecture diversity, which may influence transcriptional complexity and alternative splicing. In ORFs, Intron-Exons are those main structural body of a gene which has a pivotal role in making a functional protein. An online database, Gene Structure Display Server (GSDS) (http://gsds.cbi.pku.edu.cn/) was used for the identification of intron exons. Genomic and CDS sequences from Kiwifruit PanGenome Database were used to conduct intron exons analysis.

**Chromosomal mapping and evolutionary analysis of *NPR1* gene in *A. deliciosa*.** Chromosomal mapping and evolutionary (Ka/Ks) analysis helped trace gene duplication history and selective pressure shaping gene family expansion. The Gene Location Visualize (Advance) in TBtool was used to visualize the location of gene on chromosomes and their interactions. The data required for this analysis was Chromosomal length (retrieved from Kiwifruit PanGenome Database), Start and End position of gene (retrieved from Kiwifruit PanGenome Database), Location of Gene on chromosome (Retrieved from Kiwifruit PanGenome Database) and Gene Pair file (prepared by using OpenAI tool).

The rate of molecular evolution and gene duplication of these *NPR1* genes were calculated by Ka_Ks calculation program in TBtools. The gene pair file and CDS sequences retrieved from Kiwifruit PanGenome Database were used. The resultant file was further calculated to the MYA (Million Years Ago) value of a gene to estimate the time of evolution and duplication in these genes. The formula ($T = Ks/2\lambda$) was used to calculate the time. The value of $\lambda$ for kiwifruit was taken from supplementary file of article which was 0.00339 substitution per site per million years [17]. After the calculation, the heatmap was created by using TBtool with log 1.0 value.

**Single synteny and dual synteny analysis of *NPR1* gene in *A. deliciosa*.** Synteny analysis was conducted to explore genomic collinearity and duplication events within *A. deliciosa* and across species. For the identification of linkage of genes with interspecies and intraspecies, the dual synteny and single synteny analysis were used. For single synteny, 3 files were used in advance circos feature of TBtool to identify the linkage of genes on the chromosomes of *A. deliciosa* which describe the interaction of orthologous and paralogous linkages of genes as done by [2]

For the identification of linkage of genes on chromosomes of *A. deliciosa* with other species, Dual synteny analysis was used. For this analysis, genomic. fa file and.gff files of 4 different species (*A. thaliana, O. sativa, T. cocoa, and HongyangV4.hap2*) were used in MC ScanX program of TBtool. Three files from MC ScanX (.CTL,. COLLINEARITY, and. GFF) files were further used in Dual Synteny Plot (MC ScanX) to visualize the duplicated genes and their interactions with genes of other species.

**Transcriptome analysis of *NPR1* gene in *A. deliciosa*.** Transcriptome expression profiling was used to assess gene expression changes under fungal and bacterial stress, revealing potential functional roles in defense. To investigate the effect of *NPR1* gene in *A. deliciosa,* Transcriptome profiling data on Kiwifruit PanGenome Database was used. Data of 9 fungal (*B. cinerea*) treatments on fruit, 4 bacterial (*Pseudomonas syringae pv. Actinidiae*) treatments on leaf were used against 10 No treated fruit samples and 4 No treated leaf samples to evaluate the regulation of *NPR1* gene in *A. deliciosa* (MeiweiW1). The last 4 leaf samples were also inoculated with *Pseudomonas syringae pv. Actinidiae.* The FPKM value of identified genes in *A. deliciosa* against respective SRA ID were recorded and further used to make heatmap in TBtool. For fruit infection, kiwifruit samples were infected by necrotrophic fungal pathogen *B. cinerea* at 12h, 24h, and 48 h. and for controlled treatments, Transcriptomic data set from different kiwi plant tissues shoots (E1), leaves (E2), flower buds (E3), flowers (E4), fruits 7 days (E5), fruits 50 days (E6), fruits 120 days (E7), and fruits 160 days after full bloom.

For leaf infection, kiwifruit (*Actinidia Lind*.) Jinkui is inoculated by *Pseudomonas syringae pv. Actinidiae.* (https://www.ncbi.nlm.nih.gov/sra/SRX12358426[accn]) And for controlled treatments, Transcriptomic data set from different kiwi plant tissues shoots (E1), leaves (E2), flower buds (E3), flowers (E4), fruits 7 days (E5), fruits 50 days (E6), fruits 120 days (E7), and fruits 160 days after full bloom.

**Gene ontology and P-P Interaction of *NPR1* gene in *A. deliciosa*.** GO enrichment and protein–protein interaction analysis were conducted to link AdNPRs to specific biological processes, molecular functions, and signaling networks. Gene enrichment analysis was conducted by using Kiwifruit PanGenome Database (https://kiwifruitgenome.atcgn.com/). The biological and molecular processes of identified *NPR1* genes in *A. deliciosa* were prepared by using bar-plot with log-10 value.

Protein-to-Protein Interaction (PPI) analysis was conducted by an online database STRING v11.0 (https://string-db.org/) was used with a high confidence score of 0.7 and PPI enrichment value 9.29e-14, sourced from text mining, studies, gene fusion data, databases, and co-expression analysis [15].

## Results

### NPR1 sequence retrieval from database and BLAST-P in *Actinidia deliciosa*

NPR1 sequence retrieved from NCBI was used for motif finder resulting in the identification of the sequence for three domains, NPR1 like C domain (Des: PF12313, NPR1/NIM1 like defense protein C terminal), BTB domain (Des: PF00651, BTB/POZ domain) and Ankyrin repeats (Des: PF12796, Ankyrin repeats). The validated sequence was further used for BLAST-P program at Kiwifruit PanGenome Database (https://kiwifruitgenome.atcgn.com/) and TBtool. Both programs resulted in 8 identified hits. Peptide sequences of all identified hits were revalidated by using motif finder which resulted in 5 identified genes with all three main domains, NPR1 like C, BTB and Ank-repeats, leaving 2 identified genes which were not having NPR1 like C domain. One identified gene was duplication of one of two identified genes of no having NPR1 like C domain. Further, Kiwifruit PanGenome Database also identified these 3 hits as weaker hits.

Identified genes were renamed as *AdNPR1, AdNPR2, AdNPR3, AdNPR4*, and *AdNPR5* for further analysis of genes.

### NCBI-CDD (Conserved Domain Database) analysis

The conserved domain analysis of identified peptide sequence was conducted by NCBI-CDD resulted in the identification of 4 domains. NPR1_like_C Superfamily, BTB_POZ superfamily, ANKR superfamily and DUF3421 superfamily were found in *AdNPR1, AdNPR2, AdNPR3, AdNPR4*, and *AdNPR5* but *AdNPR6* and *AdNPR7* had no functional units of *NPR1* gene as shown in Fig 1. The post-translated protein had no respective function against biotic stress. DUF3421 (Domain of Unknown Function 3421) is a conserved protein domain that has no clearly defined or universally agreed biological function in plants. Recent research has shed light on its potential roles, especially in stress responses, signaling, and development [18].

### Phylogeny analysis of *AdNPRs* gene in *A. deliciosa* with of different species

To assess the homologs of *NPR1* gene, phylogenetic tree was constructed with 5 identified genes of *AdNPRs* in *A. deliciosa* and other major & closely related species by using MEGA11 tool and cladding was done on iTOL website. Phylogenetic Tree Fig 2 was classified into 4 major clades based on the presence of *AdNPRs* with *AtNPRs* (*A. thaliana*) as done in [2,15,16]. In Clade I, the presence of *AdNPR3, AdNPR4* and *AdNPR5* with *AtNPR3* and *AtNPR7* describes the close relationship between *NPR* genes of *A. deliciosa* and *A. thaliana.* In clade II, the presence of *AdNPR1* and *AdNPR2* revealed the resemblance and relationship between *AtNPR2* and *AtNPR4.* The remaining *AtNPRs* have no direct relationship with *AdNPR1, AdNPR2, AdNPR3, AdNPR4*, and *AdNPR5*. The presence of these *AdNPRs* with other major and closely related species indicates the functional similarity of these genes among the different species [19].

### Physio-chemical properties and Subcellular Localization of *AdNPRs* genes in *A. deliciosa*

Physio-chemical properties of *AdNPRs* genes identified by using Expasy Protparam tool resulted in characteristics of every gene as shown in accession Table 1 (Supplementary Material S1 Table in S1 File). The length of genes in *A. deliciosa* varied between 586−593 amino acids, and their molecular weight varied between 65712.82 KDa to 65832.35 KDa. Isoelectric point (pI) value varied from 5.54 to 6.36 and Instability Index (II) value varied from 43.06 to 51.04. The Aliphatic Index (AI) value varied from 89.56 to 96.54 and all GRAVY Values are negative (-ve) ranged from −0.126 to −0.253.

Subcellular Localization heatmap generated from the data retrieved from WoLF PSORT showed the location of relevant genes in organelle of cells. Localization is an important step to understand the functionality of the gene by its concentration. Fig 3 shows the presence of *AdNPR1* and *AdNPR2* in cytoplasm in high concentrations, but others were found in low concentrations in cytoplasm. *AdNPR5* is found abundantly in nucleus, but others were found in low concentrations in nucleus. *AdNPR4* and *AdNPR5* were also found in nuclear-cytoplasm.

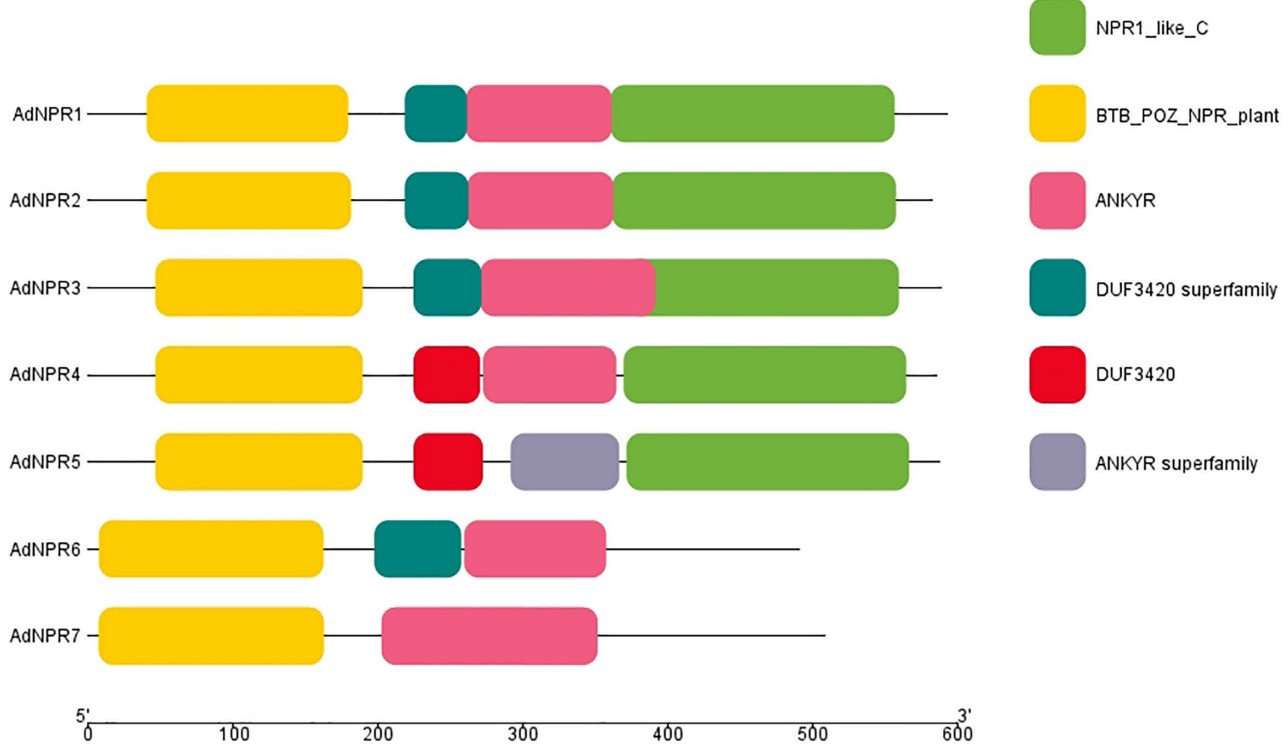

**Fig 1. NCBI-CDD Domain structures of AdNPR genes: Showing the presence of NPR1_like_C Superfamily, BTB_POZ superfamily, ANKR superfamily and DUF3421 superfamily in identified genes.**

## Identification and characterization of Gene Structure of *AdNPRs* in *A. deliciosa*

**Cis-Regulatory elements (CREs) Identification of *AdNPRs* in *A. deliciosa*.** As the most important role player of gene regulation, Cis regulatory elements have a very critical role for the activation/deactivation, upregulation/downregulation of genes. It helps to identify the function and behavior of a gene to help them to bind the transcription factors (TFs) to initiate the transcription. The query for PLANT CARE resulted in the identification of different CREs in identified genes to sort them out with their functions as shown in Fig 4. *AdNPR2* has the highest number of TATA-boxes in its promotor region as it serves as a recognition site for the binding of the TATA-binding protein (TBP) and other general transcription factors. Respectively, CAAT-Box was found abundantly in all genes which significantly boosts the efficiency of transcription.

he CAAT-box and TATA-box are classical promoter elements located near the transcription start site, playing vital roles in the initiation of transcription by facilitating the binding of transcription factors and RNA polymerase. [16] The Box-4 is part of a conserved DNA module involved in light responsiveness, as are the Sp1, G-box, and GT1-motif, all contributing to gene activation in response to light. [15] Elements like the CARE motif are involved in plant defense mechanisms, likely activated during pathogen attack or environmental stress. The ABRE motif responds to abscisic acid, a hormone essential for stress responses such as drought [2,17]. The STRE element is also linked to stress response pathways, working through interactions with specific transcription factors under stress conditions.

The motif analysis using MEME Suite revealed a conserved and structured motif architecture among the five *AdNPRs* proteins (*AdNPR1* to *AdNPR5*), indicating evolutionary conservation and potentially similar functional roles. A total of 20 motifs were identified, with the distribution and order being largely conserved, especially among closely related paralogs

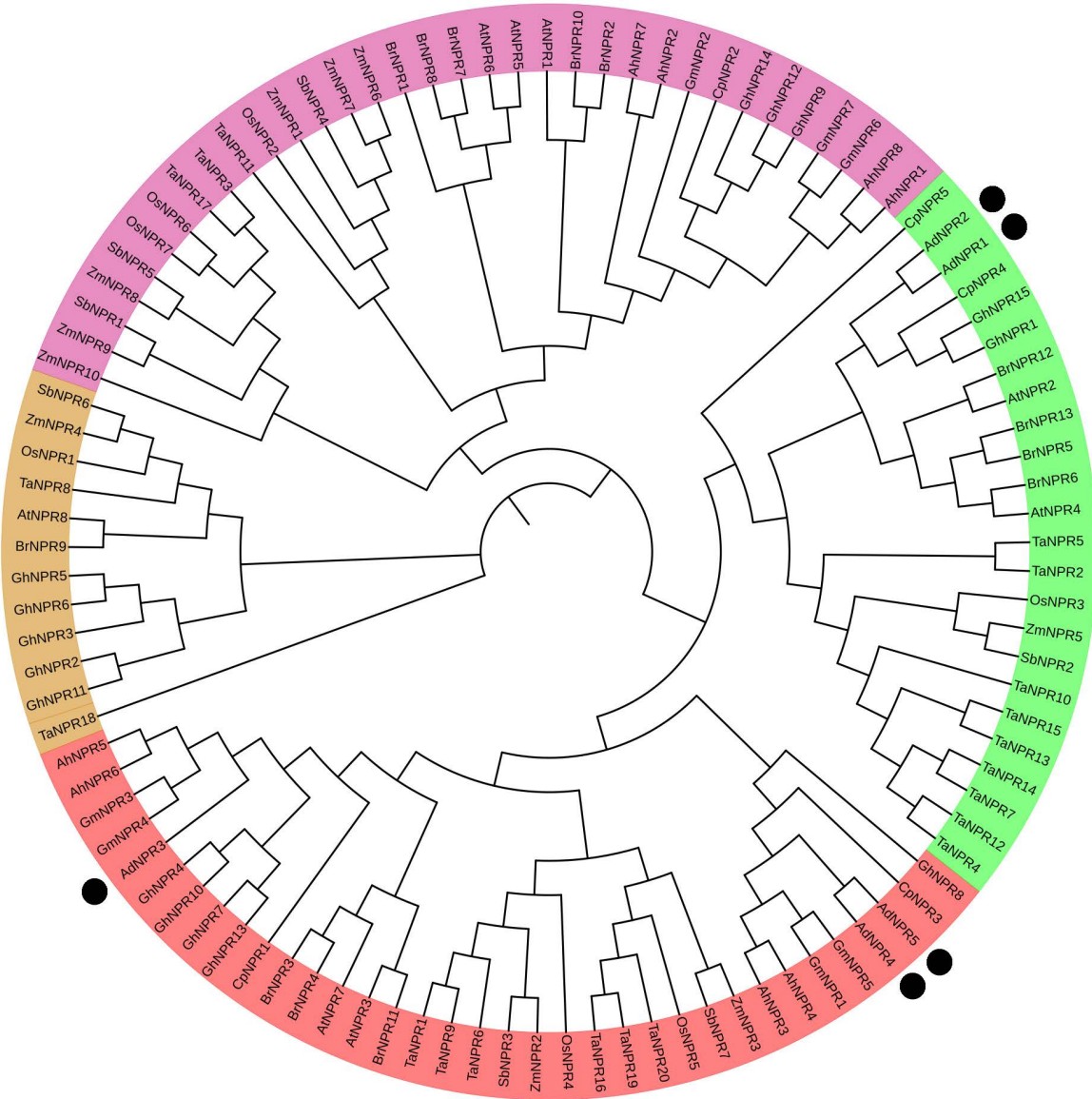

**Fig 2. Phylogenetic relationship among Arabidopsis (A. thaliana), Wheat (T. aestivum), Rice (O.** Sativa), Peanut (A. Hypogea), Sorghum (S. bicolor), Cotton (G. hirsutum), Maize (Zea mays), Canola (B. rapa), Soybean (G. max), and Papaya (C. papaya). Black dots are used to identify A. deliciosa. In iTOL, evolutionary analysis was carried out and further edited with Canva. Maximum-likelihood phylogeny of NPR1 homologs from 11 species. Black dots highlight A. deliciosa sequences; bootstrap values (>70%) are shown at nodes." Legend explains dot symbol.

such as *AdNPR1* and *AdNPR2* as shown in Fig 5. Key motifs, including Motif 1, 3, 5, and 6, were universally present across all *AdNPRs* proteins. These motifs are functionally significant as they correspond to core domains essential for NPR1 activity, particularly the BTB/POZ domain (motif 1 and 3), which facilitates protein-protein interactions and nuclear translocation during defense signaling, and the ankyrin repeat region (motif 5 and 6), known to mediate binding with TGA transcription factors. These interactions are critical for the transcriptional activation of pathogenesis-related (PR) genes. The presence of motif 17 exclusively in *AdNPR1* and *AdNPR2*, and not in the others, may suggest a specialized regulatory sub-function or recent duplication event. [15,16].

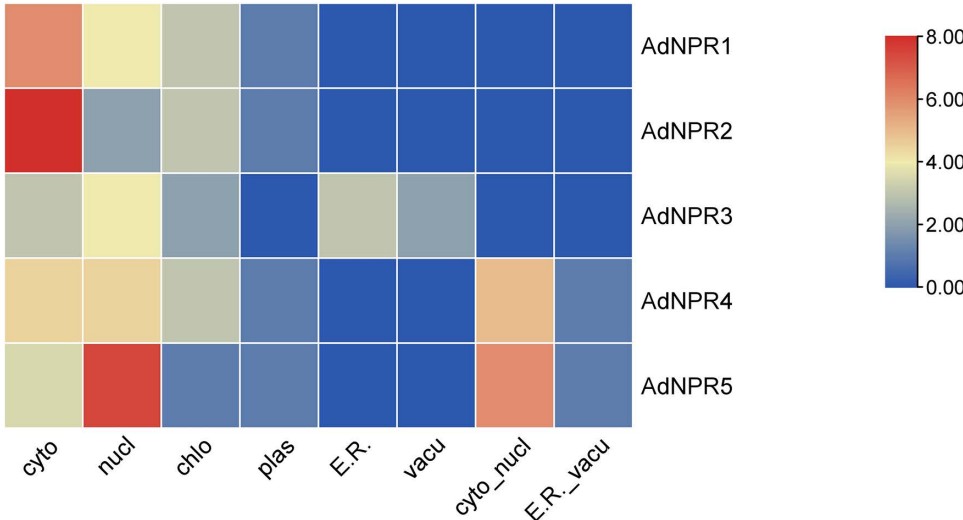

**Fig 3. Subcellular Localization: Heatmap of location of all AdNPRs genes in cells of A.** Deliciosa showing the concentration of that gene in specific organelle..

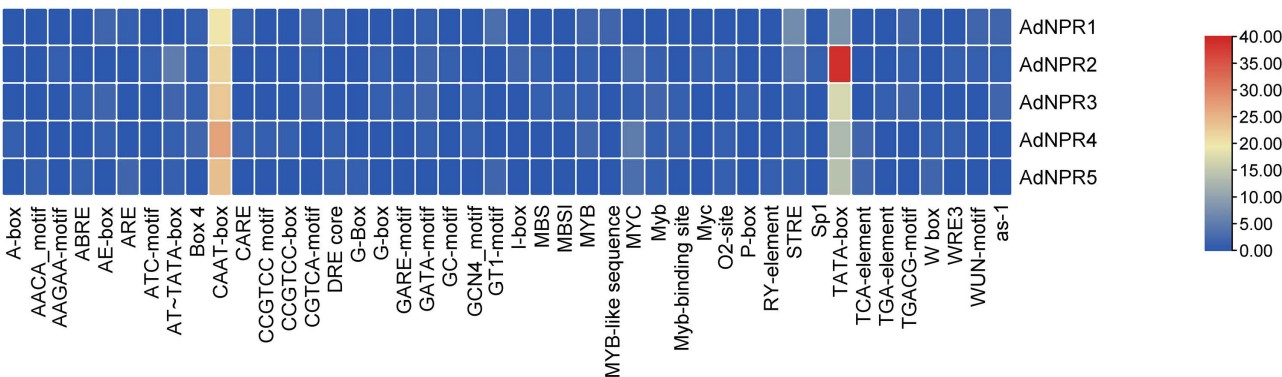

**Fig 4. Cis-Regulatory Elements:(a) Heatmap of numbers of CREs presence in promotor region of every gene.** Maximum Numbers are of TATA-box and CAAT-box.Motif and Intron Exon analysis *AdNPRs* in *A. deliciosa.*

The exon-intron structure of *AdNPRs* genes was assessed to gain insight into their genomic and evolutionary dynamics. The genes display variation in intron-exon architecture, with exon counts ranging from 3 to 8. *AdNPR1* and *AdNPR2* contain the highest number of exons, indicative of complex transcriptional regulation and possibly alternative splicing events as in Fig 6. The variation in intron lengths and positioning also suggests evolutionary divergence and potential gene duplication events, which may have facilitated the functional diversification of NPR1-like genes in *Actinidia deliciosa*. The consistency of exon-rich regions among *AdNPR1* and *AdNPR2* aligns with their close phylogenetic relationship and highly conserved domain architectures. These structural patterns are consistent with those reported in rice and sunflower *NPR1* homologs [2,15] where gene structure diversity was associated with species-specific adaptation to environmental stresses. Intron–exon variation can influence gene expression by enabling alternative splicing, which allows a single gene to produce multiple protein isoforms with distinct functions. This mechanism is often triggered under stress conditions, enabling plants to rapidly adjust gene function in response to environmental cues or pathogen attacks.

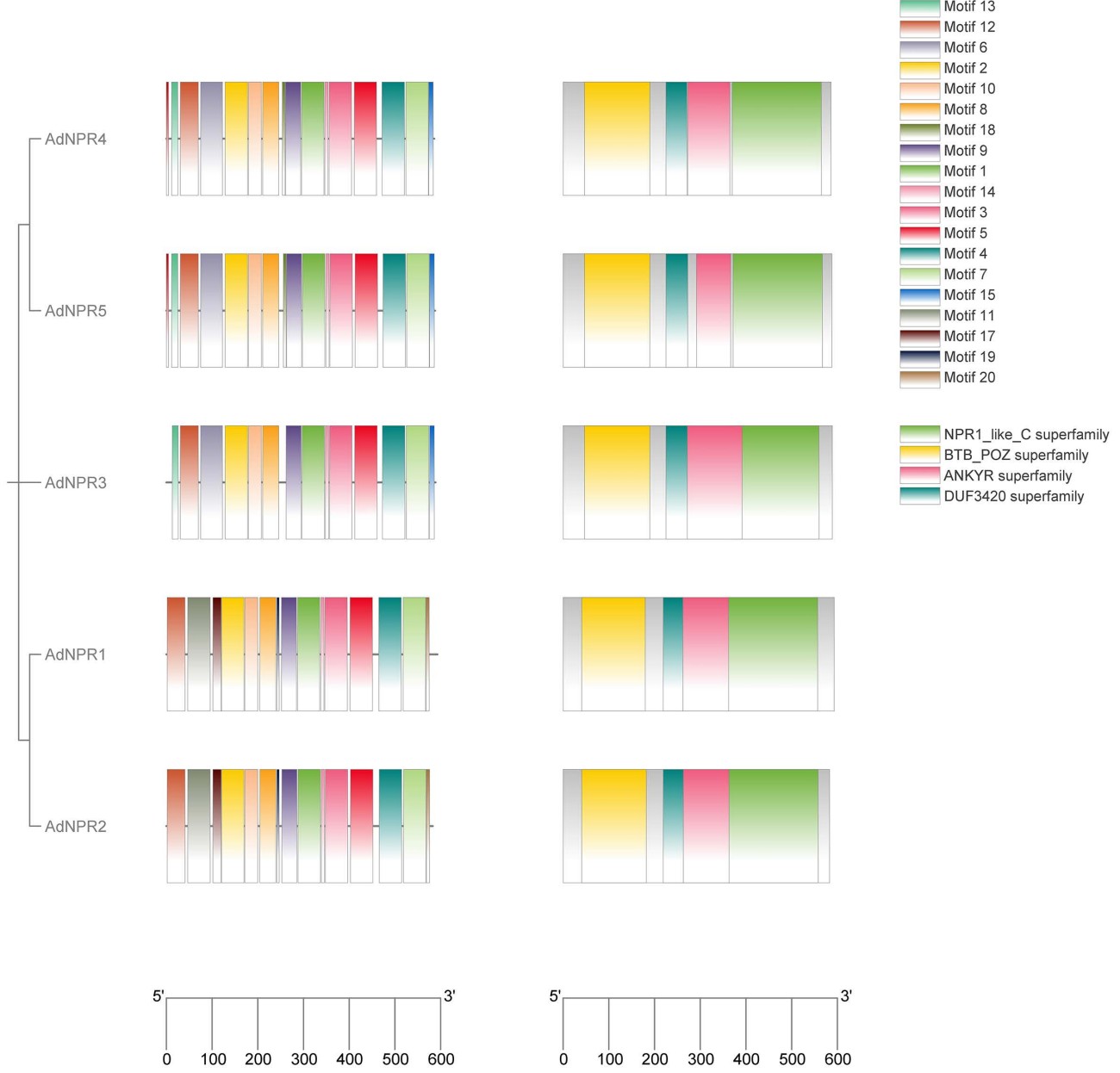

**Fig 5. Motif (a) and NCBI-CDD(b): (a) 20 motifs present with the phylogeny of AdNPRs genes in A.** Deliciosa showing the length of each motif in gene. Conserved motif 1 and motif 5 align with the ankyrin and BTB domains, highlighting their importance in NPR1 structural integrity **(b)** The NCBI-CDD showing the presence of necessary functional units of NPR1genes.

**Chromosomal mapping and evolutionary analysis of *AdNPRs* gene in *A. deliciosa*.** Chromosomal mapping is a way to visualize the location of genes along with their interaction with each other. *AdNPRs* genes were found on chromosome no. 1, 9, 14, 19, and 23. No gene was found on Chromosome no. 2, 3, 4, 5, 6, 7, 8, 10, 11, 12, 13, 15, 16, 17, 18, 20, 21, 22, 24, 25, 26, 27, 28, and 29. These 5 genes were found on 5 different chromosomes, one gene on one chromosome as in Fig 7

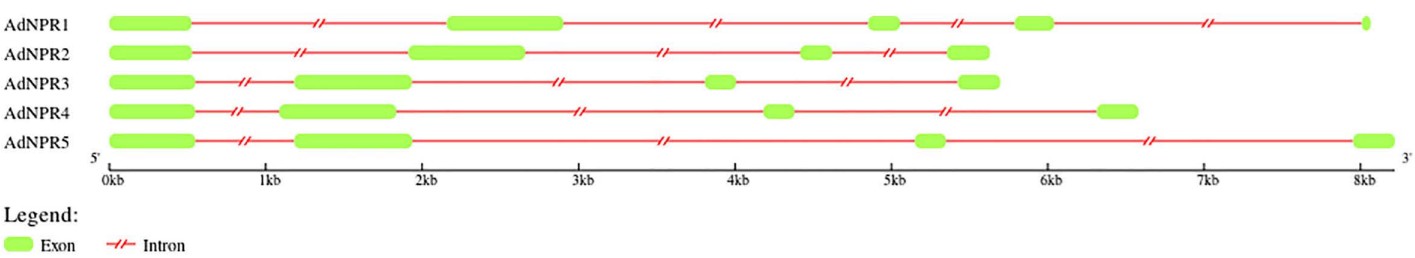

**Fig 6. Intron-Exon Analysis: graphical image showing the presence of numbers of introns and exons of a gene indicating the function of gene.**

*AdNPR2_AdNPR5* had the highest divergence time (625 MYA) and a low Ka/Ks ratio (0.132), suggesting early divergence with functional constraint [17]. *AdNPR3_AdNPR4* and *AdNPR3_AdNPR5* also diverged early (209–219 MYA), maintaining low Ka/Ks values (0.194–0.211) as in Fig 8 In contrast, *AdNPR1_AdNPR2* (28.9 MYA) and *AdNPR4_AdNPR5* (25.2 MYA) represent recent duplications yet still show strong purifying selection (Ka/Ks ~ 0.20–0.25). This suggests that despite divergence times, all gene pairs retained essential functional roles [15]. The highest divergence time (~625 MYA) between *AdNPR2* and *AdNPR5* likely corresponds to early angiosperm lineage diversification, [17] indicating that the *NPR1* gene family is ancient and evolutionarily conserved. The consistently low Ka/Ks ratios across gene pairs suggest that purifying selection has maintained the functional integrity of these genes over time [2]

**Single synteny and dual synteny analysis of *AdNPRs* gene in *A. deliciosa*.** Duplication is the most fundamental biological process in evolution of an organism, playing a vital role in producing functional variation and advancement in an organism. Duplication process starts from the chromosomes of one organism and then adapted by the environment. To study this duplication process and locus relationship of *AdNPR1* genes in *A. deliciosa,* single synteny analysis was conducted in TBtool. We identified the paralogous gene. The five identified *AdNPRs* genes are unevenly dispersed across six chromosomes, Chr1, Chr6, Chr9, Chr14, Chr19, and Chr23. Each gene is distinctly labeled on its respective chromosome, showing a non-clustered arrangement. Syntenic relationships among these genes are indicated by colored connecting lines, highlighting segmental duplication events [2,16]. The strong synteny is observed between gene pairs on Chr1-Chr19, Chr9-Chr23, and Chr14-Chr6, suggesting that gene duplication has significantly contributed to the expansion of the *NPR1*-like gene family in Kiwifruit as in Fig 9

In the *A. deliciosa* vs *A. thaliana* comparison, conserved gene blocks were observed between chromosomes such as Chr02 and Chr03 of *A. thaliana* and Chr03, Chr10, and Chr25 of *A. deliciosa,* suggesting evolutionary conservation despite species divergence. Similarly, in Fig 10 the A. *deliciosa* vs *T. cacao* synteny displayed clear orthologous regions, particularly between Chr01, Chr04, and Chr09 of *A. deliciosa* and Chromosomes 1, 4, 5, and 9 of T. cacao. This aligns with findings by [2,20] which highlighted the ancient and functionally conserved nature of *NPR1*-like genes across dicots. There was no linkage of *NPR1* genes found between *A. deliciosa* and *A. chinensis* and *O. sativa*

**Transcriptome analysis of *AdNPRs* gene in *A. deliciosa*.** The heatmap analysis of FPKM values revealed differential expression patterns of *AdNPRs* genes in response to fungal (B. cinerea) and bacterial (Pseudomonas syringae pv. actinidiae) treatments. *AdNPR4* showed consistently high expression under both fungal-infected fruit samples and bacterial-infected leaf samples, suggesting a central role in defense signaling across tissues. *AdNPR3* also displayed moderate to high expression in pathogen-treated samples, particularly under fungal stress, indicating a likely involvement in early stress response mechanisms. *AdNPR1* and *AdNPR5* remained weakly expressed or unaltered across most treated conditions, suggesting a lesser or more condition-specific role in defense. In Fig 11, *AdNPR2* expression was slightly upregulated under fungal treatment but remained comparatively moderate across treatments. *AdNPR3* and *AdNPR4* are transcriptionally responsive to biotic stress, especially under necrotrophic fungal attack and bacterial infection, highlighting their potential as key regulators in the immune response pathway of *Actinidia deliciosa*.

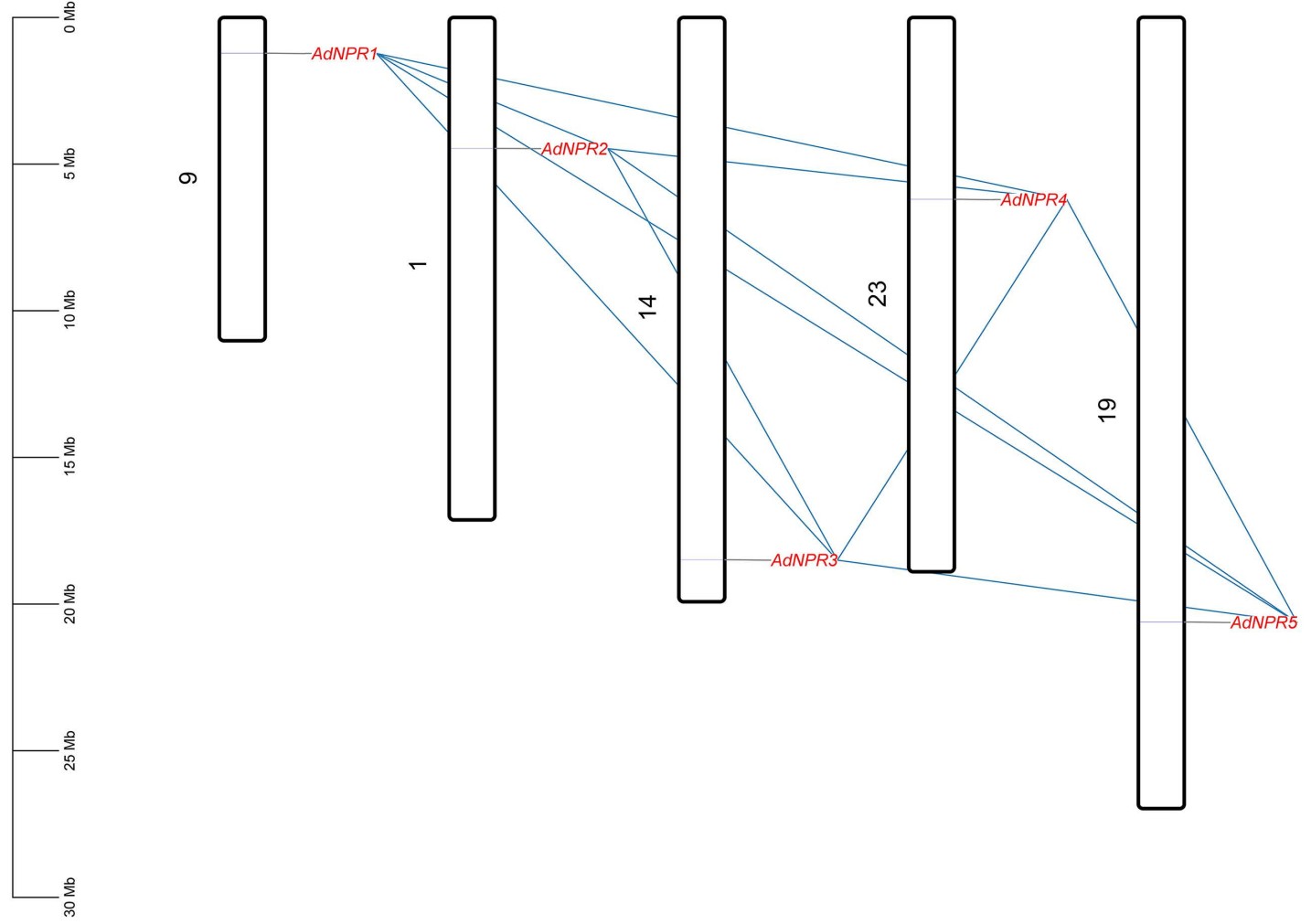

**Fig 7. Chromosomal mapping: Chromosomal locations of AdNPR genes in A. deliciosa.** Genes are located on chromosomes 1, 9, 14, 19, and 23. Black blocks with white colour inside represent the chromosome. Difference in size, present the size of chromosome. Blue lines show the linkage of genes between them.

**Gene ontology and P-P Interaction of *AdNPRs* gene in *A. deliciosa*.** GO enrichment analysis of *AdNPRs* genes in *Actinidia deliciosa* revealed their major roles in two key biological areas: developmental regulation and defense response. Development-related terms such as nectary development, stipule development, and meristem determinacy suggest involvement in floral and organ patterning. Meanwhile, enriched defense terms like systemic acquired resistance, jasmonic and salicylic acid signaling, and response to herbivore/insect highlight their critical role in biotic stress response pathways, indicating that *AdNPRs* genes coordinate growth and immunity as shown in Fig 12.

STRING-based interaction network analysis revealed that the NPR1-like proteins in Actinidia chinensis form a tightly connected cluster of five nodes with six edges, exhibiting significantly higher connectivity than expected (PPI enrichment p-value = 9.29e-14). This suggests strong functional associations among these proteins as shown in Fig 13. Functional enrichment analysis indicated their involvement in key defense-related biological processes of Regulation of salicylic acid-mediated signaling pathway (GO:2000031), Response to salicylic acid, fungus, and bacterium (GO:0009751, GO:0050832, GO:0042742) as in [2] and cellular responses to organic cyclic compounds (GO:0071407).

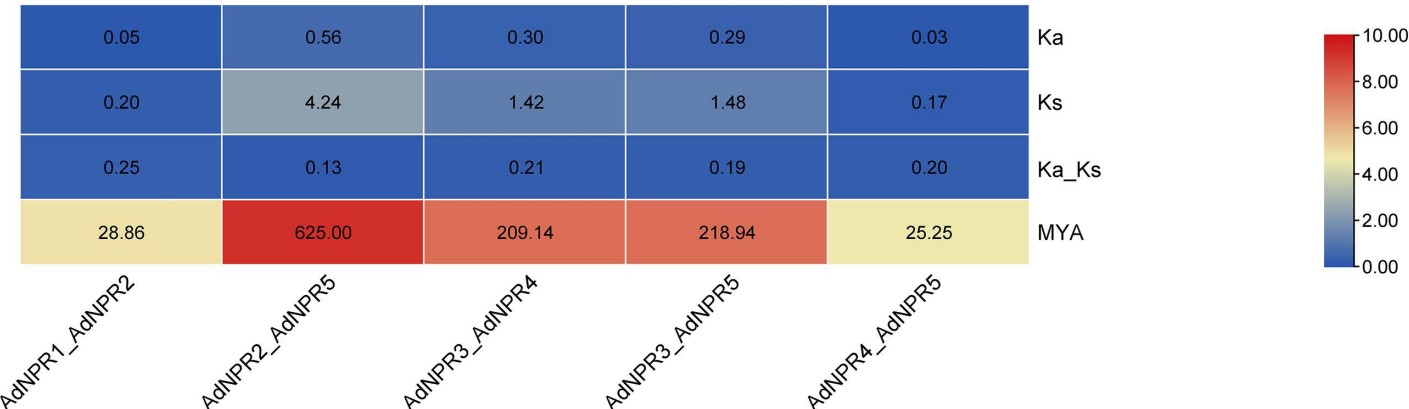

**Fig 8. Ka_Ks analysis: Heatmap showing the MYA value of gene duplication time.** AdNPR2_AdNPR5 sequence show largest MYA value indicating the 625 million years ago divergence.

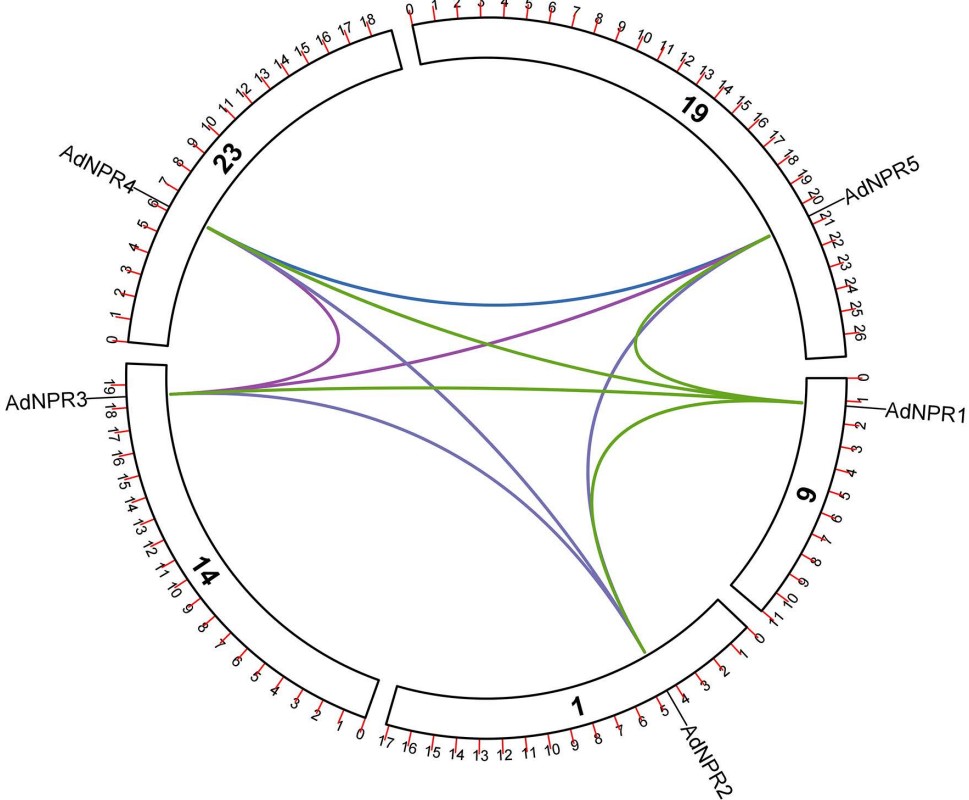

**Fig 9. Synteny Analysis: The genes are present on chr. no. 1, 9, 14, 19, 23.** Their linkage are showing only segmental duplication as all lines from one chromosome links with gene only on other chromosome. There is no tandem duplication in **A.** Deliciosa.

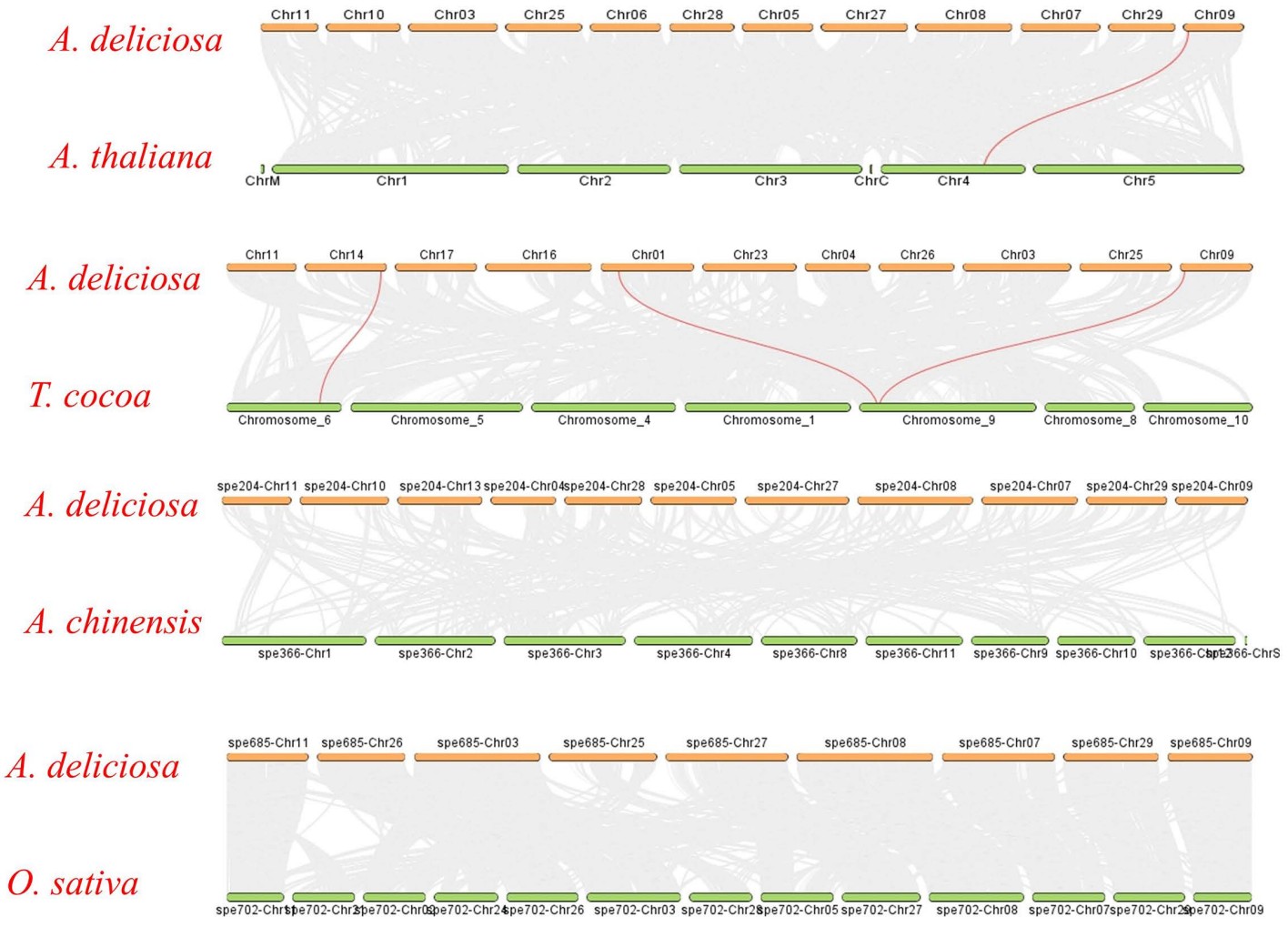

**Fig 10. Dual Synteny Analysis: the MC ScanX link the genes from A.** Deliciosa to other species as in pic. **A.** Deliciosa represent fuzzy kiwi fruit, **A.** Thalina to Arabidopsis, **A.** Chinensis to kiwi hypogea and **O.** Sativa to rice. Red line between the chromosomes shows the linkage of species with respect of NPR1 gene. No line in **A.** Chinensis and **O.** Sativa shows that there is no resemblance with them which show that gene conservation is strongest with T. cacao and A. thaliana, while absent in A. chinensis, possibly due to annotation inconsistencies.

## Discussion

The present study offers the first comprehensive genome-wide identification and characterization of *NPR1*-like genes in *Actinidia deliciosa*, unravelling their evolutionary dynamics, structural features, stress-responsive expression, and functional potential through integrated in-silico approaches. These findings not only deepen our understanding of the *NPR1* gene family's role in kiwifruit defense but also align with previously reported studies across various crop genomes, including *Theobroma cacao* [2], *Oryza sativa* [16], and *Arabidopsis thaliana* [1].

The identification of five functionally complete *AdNPRs* genes, each encoding the conserved BTB/POZ, ankyrin repeats, and NPR1-like C-terminal domains, reveals structurally preserved and functionally distinct genes [21,22]. The presence of the DUF3421 domain in some paralogs, whose function remains ambiguous, could hint at a neofunctionalization trajectory or partial domain loss. Such incomplete duplications have also been reported in tomato *NPR1*-like gene families [23], suggesting lineage-specific pressures on domain retention. While domain predictions are

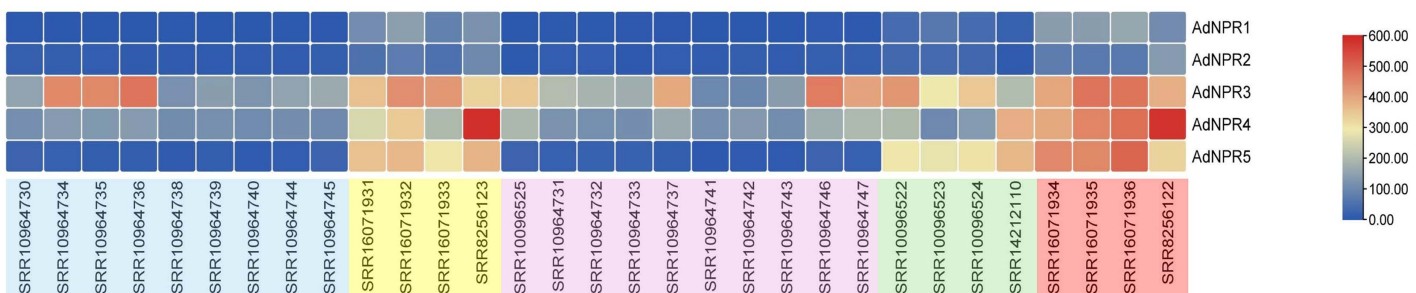

**Fig 11. Expression Analysis: The SRA Ids highlighted with light blue colour indicate the sample is of fruit with fungal (B.** Cinera), Yellow colour indicates the SRA Ids of leaf sample with bacterial treatments (Pseudomonas syringae pv. Actinidiae). Purple colour shows the SRA ids of sample of fruit with no treatment. Green shows the samples of leaf with no treatment and red shows the sample of lead with bacterial treatment (Pseudomonas syringae pv. Actinidiae). The heatmap with max. red colour shows the upregulation of the genes in response to infection.

valuable, their actual activity and protein-folding stability under stress remain to be validated, calling for structural modeling or domain-deletion mutagenesis studies [24]. Functional relevance of the DUF3421 domain remains unresolved.

The GRAVY (Grand Average of Hydropathy) value and isoelectric point (pI) of a protein provide valuable clues about its function and cellular localization. A positive GRAVY value indicates a hydrophobic nature, suggesting that the protein may be membrane-bound or associated with lipid environments, while a negative GRAVY value points to a hydrophilic and likely soluble protein. Similarly, the isoelectric point reflects the pH at which the protein has no net charge; proteins with higher pI values tend to function in acidic compartments such as lysosomes, whereas those with lower pI values are often localized in more basic environments like the cytoplasm or nucleus. Together, these parameters help infer the biochemical behavior, potential interaction partners, and subcellular targeting of proteins. [2]

Phylogenetic analysis grouped the *AdNPRs* genes into distinct clades, closely aligning with *AtNPR1/2/3/4* of Arabidopsis and TcNPR1 of *Theobroma cacao* [2], implying evolutionary conservation and potential sub-functional diversification. The diversification seen in Clade I and II, aligning with *NPRs* of both monocots and dicots, supports the notion of *NPR1*-like genes evolving from a common ancestor prior to the monocot-dicot split [25]. In (Figure 2), the tree was divided into four major clades. *AdNPR1* and *AdNPR2* grouped closely with Arabidopsis thaliana *NPR1* (*AtNPR1*) and *AtNPR2*, suggesting that these genes may retain the canonical function of activating salicylic acid-mediated systemic acquired resistance (SAR), like *AtNPR1*. This grouping implies a conserved regulatory role in basal defense activation. In contrast, *AdNPR3* and *AdNPR4* formed a separate cluster with *AtNPR3* and *AtNPR4*, known negative regulators of defense responses [15]. This suggests potential sub-functionalization of these genes in *A. deliciosa*, possibly modulating NPR1-mediated signaling to prevent hyperactivation of immune responses and balance energy allocation between growth and defense. Interestingly, *AdNPR5* formed a distant node within the same clade, indicating divergence that may reflect novel or intermediate regulatory functions, potentially linked to developmental processes or stress fine-tuning [26]. The clear separation of these clades, and their alignment with known functional orthologs in *Arabidopsis, Theobroma cacao,* and *Oryza sativa*, supports the hypothesis that *NPR1*-like genes in *A. deliciosa* have evolved through both conservation and diversification to accommodate lineage-specific adaptation [27].

CREs in promoter regions provide a mechanistic foundation for the transcriptional responsiveness of *NPR1* genes. The abundance of TATA-box, CAAT-box, ABRE, G-box, and light-responsive elements suggest that *AdNPRs* are likely influenced by both endogenous (hormonal) and exogenous (pathogen and light) cues [28]. Notably, the abundance of ABRE in *AdNPR2* hints at its probable involvement in ABA-mediated stress signaling, aligning with *NPR1's* known role in hormonal crosstalk [15,16]. The promoter regions of *AdNPRs* genes contained several hormones and stress-responsive cis-elements, including ABRE (abscisic acid) [27], CGTCA-motif (jasmonic acid), [2,29–32] TGA-element (auxin), and

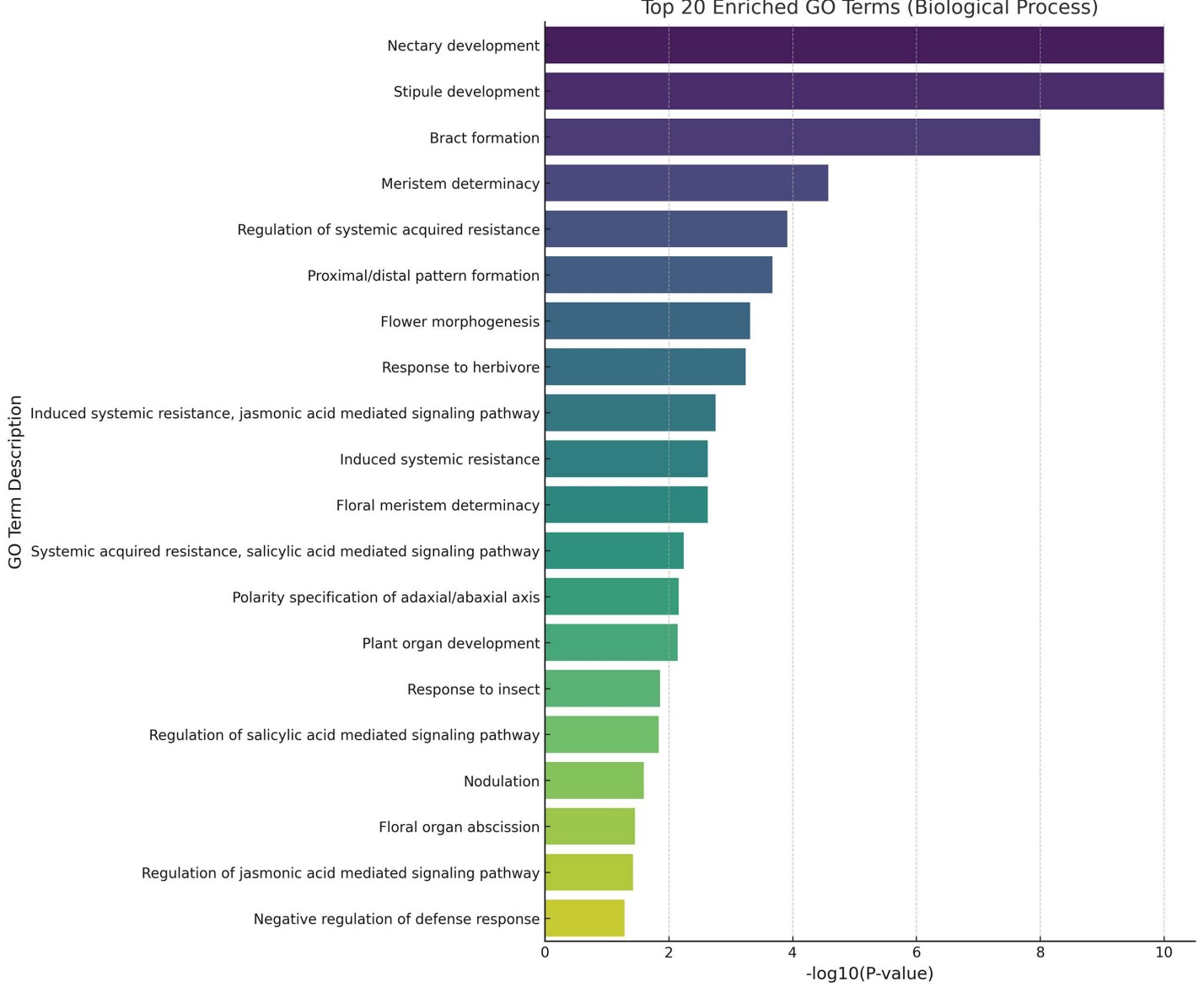

**Fig 12. GO analysis: The bar plots with p value of -log10 indicate the role of NPR genes in their molecular, biological and functional processes.** Longer the bar, maximum function performed by these genes.

W-box (WRKY binding). These elements suggest potential binding by transcription factors involved in ABA, JA, and SA signaling pathways, indicating that *AdNPRs* gene expression is likely regulated through complex hormonal and defense-related networks [2].

All 5 *AdNPRs* proteins exhibit the canonical NPR1 domain architecture, featuring four key domains: BTB/POZ superfamily domain, DUF3420 domain, ANKYR (ankyrin repeat) superfamily domain, and NPR1_like_C superfamily domain [25]. This conservation suggests that the fundamental molecular functions of NPR1 proteins are likely preserved in kiwifruit, particularly their roles in salicylic acid (SA) perception and defense response regulation [2]

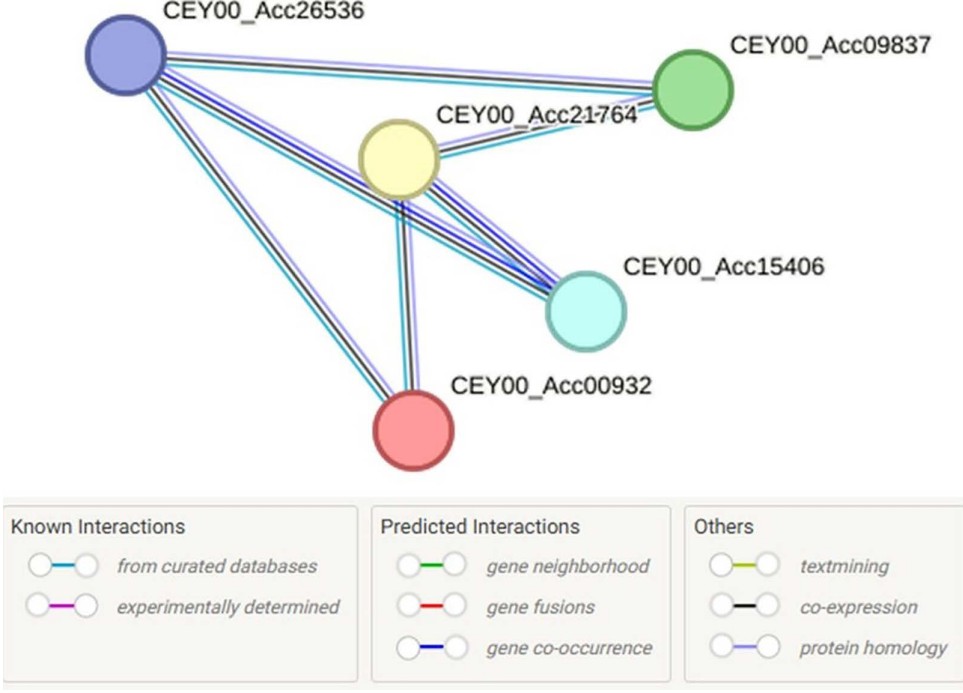

**Fig 13. Protein-to-Protein interaction: Indicates the linkage of the protein structures involved in multiple cellular pathways.**

The BTB/POZ domain (yellow in Fig. 2) is located at the N-terminal region of all *AdNPRs* proteins, consistent with its well-documented role in protein-protein interactions crucial for transcriptional regulation. This domain is particularly important for interactions with TGA transcription factors during defense response activation [21]. Adjacent to the BTB/POZ domain is the DUF3420 domain (teal in Fig. 2), whose specific function remains largely unknown but is consistently present in plant NPR proteins. The central region of each *AdNPRs* protein contains the ankyrin repeat domain (pink in Fig. 2), which mediates protein-protein interactions essential for NPR1 function in defense signaling. At the C-terminus, all *AdNPRs* proteins possess the NPR1_like_C domain (green in Fig. 2), which functions as a transactivation domain necessary for defense gene induction [16].

The detailed motif analysis revealed 20 conserved motifs (numbered 1–20) with varying distribution patterns across the five *AdNPRs* proteins. *AdNPR1* and *AdNPR2* display nearly identical motif arrangements, further supporting their classification as closely related paralogs [33]. Both proteins share a distinctive motif 17 (brown) that is absent in other family members, potentially indicating specialized functional features. Similarly, *AdNPR4* and *AdNPR5* exhibit highly similar motif patterns to each other, suggesting they may form another functionally related pair within the *AdNPRs* family [2,29]. *AdNPR3* shows a somewhat distinct motif composition compared to the other family members, particularly in the BTB/POZ domain region, which may reflect divergent evolutionary history and potentially specialized functional roles.

Chromosomal distribution and synteny analysis highlighted a segmental duplication-based origin of the *NPR1* gene family in kiwifruit. The absence of tandem duplications and the presence of syntenic blocks between Chr1–Chr19 and Chr9–Chr23 suggest that whole-genome or large-scale segmental duplications have shaped the family. This pattern echoes the conserved duplication signatures reported in dicots like Brassica rapa [25] and T. cacao [2]. The absence of detectable synteny between *A. deliciosa* and *A. chinensis,* despite their close taxonomic relationship, may be due to differences in genome annotation quality, assembly completeness, or sub-genomic divergence after speciation. Such

discrepancies have been reported in other Actinidia studies and highlight the need for more refined comparative genomic datasets for accurate ortholog identification [15,34].

The evolutionary dynamics of the *NPR1*-like gene family in *Actinidia deliciosa*, Ka (non-synonymous), Ks (synonymous) substitution rates, Ka/Ks ratios, and approximate divergence times (MYA – million years ago) were calculated for five *AdNPRs* gene pairs [35]. The Ka/Ks ratio is widely used to infer the selective pressure acting on duplicated genes, where values <1 indicate purifying selection, values = 1 suggest neutral evolution, and >1 imply positive selection [17]. The gene pair *AdNPR2_AdNPR5* exhibited the highest divergence time at approximately 625 MYA, with a relatively low Ka/Ks ratio of 0.132, indicating that although these genes diverged early, they have been maintained under strong purifying selection with minimal functional divergence [2]. Similarly, *AdNPR3_AdNPR5* and *AdNPR3_AdNPR4* show Ka/Ks values of 0.194 and 0.211, respectively, along with divergence times of 218.9 MYA and 209.1 MYA, suggesting that these genes diverged from a common ancestor a long time ago and have since experienced moderate purifying selection [36]. The gene pair *AdNPR1_AdNPR2* recorded a Ka/Ks ratio of 0.248, reflecting strong purifying selection pressure. Its relatively lower MYA value of 28.86 suggests that this divergence occurred more recently compared to the others, possibly because of a more recent duplication event [37]. The *AdNPR4_AdNPR5* pair presented the lowest Ks value and an MYA of 25.25, indicating the most recent divergence event among the studied gene pairs [17]. Despite the young divergence age, the Ka/Ks ratio remains low at 0.198, reaffirming that the evolutionary trajectory has favored the conservation of protein function under purifying selection.

The differential expression of *AdNPRs* genes under Botrytis cinerea and Pseudomonas syringae infection suggests tissue-specific and pathogen-specific activation [19]. The strong upregulation of *AdNPR3* and *AdNPR4* across both fruit and leaf tissues implies their central role in SA-mediated defense signaling [28]. Meanwhile, *AdNPR1* and *AdNPR5* remained mostly unresponsive, indicating potential non-involvement or delayed response under the tested conditions. These expression trends reflect similar functional divergence observed in *NPR1–NPR3* expression cascades of Arabidopsis and *Solanum lycopersicum* during pathogen attack [1,23].

GO and STRING analyses underscored the bifunctionality of *AdNPRs*, participating in both developmental regulation (floral meristem determinacy, nectary formation) and defense responses (SAR, JA/SA signaling) [23]. Such dual-functional gene networks are rarely studied in fruit crops yet offer insights into how plants balance immune readiness with reproductive success [2]. STRING network analysis revealed interactions among AdNPRs proteins and components of the salicylic acid pathway, including predicted associations with TGA transcription factors and *NPR3*-like repressors as studied in [2,19,25,38]. These align with known *NPR1* signaling pathways. Notably, the predicted interaction between *AdNPR4* and a hypothetical JA-related interactor may represent a novel regulatory connection, warranting further validation. The predicted protein–protein interaction network also shows that these proteins might act in complexes, possibly with TGA transcription factors or SA receptors, to fine-tune downstream PR gene expression [9].

## Conclusion

In this study, five *NPR1*-like genes were identified from the whole genome of *Actinidia deliciosa*. Based on promoter analysis, cis-regulatory elements were found to be associated with salicylic acid responsiveness, abscisic acid responsiveness, biotic stress response, and light regulation, suggesting their involvement in hormone-mediated and environmental signaling pathways. Structural analysis revealed that the number of exons ranged from three to eight, with conserved motif patterns across all genes. RNA-Seq data analysis indicated that *AdNPR3* and *AdNPR4* showed significant upregulation under Botrytis cinerea and Pseudomonas syringae pv. actinidiae infections, highlighting their putative role in defense activation. These expression changes imply the regulatory role of *NPR1*-like genes in the salicylic acid signaling pathway and their potential function in enhancing immune responses in kiwifruit. However, further studies including gene cloning, functional validation, and mutant analysis are required to confirm the roles of these genes in plant immunity and developmental regulation.

## Limitations and future validations

This study presents a foundational genomic analysis of the *NPR1* gene family in *A. deliciosa.* However, functional validation through wet-lab techniques such as qRT-PCR, CRISPR-Cas9–mediated knockout studies, and subcellular localization assays is essential to confirm the predicted roles. Future research should also investigate protein–protein interaction dynamics and the effects of overexpression on plant immunity and development [39].

## Supporting information

**S1 File. Supplementary File.**
(DOCX)

## Acknowledgments

The author acknowledged the present study to Guizhou Provincial Major Scientific and Technological Program. Key Laboratory of kiwifruit resources development and utilization of Guizhou Universities. Liupanshui Normal University and the Science and Technology department of Liupanshui City.

## Author contributions

**Conceptualization:** Weimin Zhong, Yuexia Wang, Shiming Han.

**Data curation:** Weimin Zhong, Yuexia Wang, Shiming Han, Jihong Dong, Yumei Fang, Muhammad Umar Rasheed, Aiman Malik, Qurban Ali, Jia Zhou.

**Formal analysis:** Weimin Zhong, Jihong Dong, Xiaoling Xu, Muhammad Umar Rasheed, Aiman Malik, Qurban Ali, Muhammad Ashfaq, Jia Zhou.

**Funding acquisition:** Yumei Fang, Xiaoling Xu.

**Investigation:** Weimin Zhong, Yuexia Wang, Shiming Han, Yumei Fang, Xiaoling Xu, Jia Zhou.

**Methodology:** Yuexia Wang, Jihong Dong, Yumei Fang, Xiaoling Xu, Muhammad Umar Rasheed, Aiman Malik, Qurban Ali, Jia Zhou.

**Project administration:** Shiming Han.

**Resources:** Weimin Zhong, Yuexia Wang, Jihong Dong, Yumei Fang, Xiaoling Xu, Muhammad Umar Rasheed, Aiman Malik, Qurban Ali, Muhammad Ashfaq.

**Software:** Jihong Dong, Yumei Fang, Xiaoling Xu, Muhammad Umar Rasheed, Aiman Malik, Muhammad Ashfaq, Jia Zhou.

**Supervision:** Yuexia Wang, Shiming Han, Xiaoling Xu.

**Validation:** Weimin Zhong, Yumei Fang, Muhammad Umar Rasheed, Qurban Ali, Muhammad Ashfaq.

**Visualization:** Shiming Han, Jihong Dong, Yumei Fang.

**Writing – original draft:** Weimin Zhong, Yuexia Wang, Shiming Han.

**Writing – review & editing:** Qurban Ali, Muhammad Ashfaq, Jia Zhou.

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
