## [Decision Letter · Decision Letter 0]

28 May 2025

Dear Dr. Ali,

Thank you for submitting your manuscript to PLOS ONE. After careful consideration, we feel that it has merit but does not fully meet PLOS ONE’s publication criteria as it currently stands. Therefore, we invite you to submit a revised version of the manuscript that addresses the points raised during the review process.

We look forward to receiving your revised manuscript.

Kind regards,

Abhijeet Shankar Kashyap

Academic Editor

PLOS ONE

Journal Requirements:

“Key Laboratory of kiwifruit resources development and utilization of Guizhou Universities (Qian Jiaoji [2022] 054) ; Project of Liupanshui Normal University(No.LPSSYKYJJ201601; LPSSY2023XKTD09)and the Science and Technology project of Liupanshui City (Grant #52020-2020-0906).”

“Key Laboratory of kiwifruit resources development and utilization of Guizhou Universities (Qian Jiaoji [2022] 054) ; Project of Liupanshui Normal University(No.LPSSYKYJJ201601; LPSSY2023XKTD09)and the Science and Technology project of Liupanshui City (Grant #52020-2020-0906).”

“Key Laboratory of kiwifruit resources development and utilization of Guizhou Universities (Qian Jiaoji [2022] 054) ; Project of Liupanshui Normal University(No.LPSSYKYJJ201601; LPSSY2023XKTD09)and the Science and Technology project of Liupanshui City (Grant #52020-2020-0906).”

Reviewers' comments:

Reviewer's Responses to Questions

**Comments to the Author**

1. Is the manuscript technically sound, and do the data support the conclusions?

Reviewer #1: Yes

Reviewer #2: Yes

2. Has the statistical analysis been performed appropriately and rigorously?

Reviewer #1: No

Reviewer #2: Yes

3. Have the authors made all data underlying the findings in their manuscript fully available?

Reviewer #1: Yes

Reviewer #2: Yes

4. Is the manuscript presented in an intelligible fashion and written in standard English?

Reviewer #1: Yes

Reviewer #2: Yes

Reviewer #1: This manuscript explores the genome-wide identification and characterization of NPR1-like genes in Actinidia deliciosa, aiming to understand their structural, evolutionary, and functional roles in plant defense. The topic is timely and relevant for crop improvement under biotic stress. The bioinformatics analyses are comprehensive, and the manuscript provides substantial insights. However, the manuscript requires several corrections and improvements to language, formatting, and clarity of scientific presentation.

Reviewer #2: The manuscript Genome-Wide Identification and Functional Characterization of NPR1-Like Genes in Actinidia deliciosa has been written but needs some modifications as below

1. "five candidate genes (AdNPR1 – AdNPR5)" - It would be good to add here that they contain conserved BTB/POZ and ankyrin repeat domains, as this is a key finding (in abstract)

2. "Rasheed et al., 2025" - Please check all the publication years and ensure they are correct. There are quite a few 2025 citations.

3. "Of these, gray mold caused by Botrytis cinerea is one of the most devastating diseases." - It might be helpful to briefly mention why gray mold is so devastating (e.g., yield loss, postharvest decay).

* Materials and Methods:

1. "Kiwifruit PanGenome Database (https://kiwifruitgenome.atcgn.com/) was used for BLAST-P program on 16th April 2025 at 18:23 Pakistan (UTC+5)" - While specific details are good, the time and date of the search might not be necessary. Consider removing it for brevity.

2. "Gene Pair file (prepared by using OpenAI tool)" - This is interesting, but it needs more explanation. How was OpenAI used to prepare this file? What kind of data was input, and what was the output?

* Results:

1. Figure 1 NCBI-CDD results: Showing the presence of..." - Improve figure captions to be more informative. Instead of "Showing the presence of...", describe what the reader should observe in the figure (e.g., "Domain structures of AdNPR genes, showing the presence of...").

2. Black dots are used to identify A. deliciosa." - This is too simplistic. The caption should explain what the phylogenetic tree shows in more detail (e.g., "Phylogenetic relationships among NPR1 homologs in A. deliciosa and other plant species..."). Also, it says "Black dots are used to identify A. deliciosa" but the black dots in Figure 2 are not labelled.

3. "Figure 3 shows the presence of AdNPR1 and AdNPR2 in cytoplasm in high concentrations..." - Be more specific about what "high" and "low" concentrations mean. Refer to the scale in the heatmap.

4. "AdNPR2 has highest number of TATA-box in its promotor region as it serves as a recognition site..." - Rephrase for better flow: "AdNPR2 has the highest number of TATA-boxes in its promoter region, which serve as recognition sites...".

5. "Figure 7 Chromosomal mapping: The presence of AdNPRs genes on chromosome number1, 9, 14, 19, and 23." - Improve the caption. For example: "Chromosomal locations of AdNPR genes in A. deliciosa. Genes are located on chromosomes 1, 9, 14, 19, and 23." Also, in the caption, it says "chromosome number1" should it be "chromosome number 1"?

6. Discuss any limitations of the study and suggest potential directions for future research.

7. Domain Analysis: In Figure 1, you show the presence of different domains. Discuss the functional implications of the presence or absence of specific domains in the AdNPR genes.

8. Physicochemical Properties: While you present the physicochemical properties, discuss their biological relevance. For example, how might the GRAVY values or isoelectric points relate to protein function or localization

9. Phylogenetic Tree Interpretation: Expand on the evolutionary relationships revealed by the phylogenetic tree. Discuss the implications of the clustering patterns for the functional evolution of NPR1 genes in kiwifruit.

**Do you want your identity to be public for this peer review?** For information about this choice, including consent withdrawal, please see our Privacy Policy

Reviewer #1: **Yes: ** Muhammad Zeshan Haider

Reviewer #2: No

---

## [Author Response · Author response to Decision Letter 1]

12 Jul 2025

Reply for reviewer for Review Report for manuscript entitled:

“Genome-Wide Identification and Functional Characterization of NPR1-Like Genes in Actinidia deliciosa”

The reply/answer/response for each comment under specified number is mentioned as bulleted points with highlighted text with yellow color after the statement of comment.

Reviewer 1

General Comments

This manuscript explores the genome-wide identification and characterization of NPR1-like genes in Actinidia deliciosa, aiming to understand their structural, evolutionary, and functional roles in plant defence. The topic is timely and relevant for crop improvement under biotic stress. The bioinformatics analyses are comprehensive, and the manuscript provides substantial insights. However, the manuscript requires several corrections and improvements to language, formatting, and clarity of scientific presentation.

• We sincerely thank the reviewers and editor for recognizing the relevance and value of our study.

• In response to the general feedback, we have carefully revised the manuscript to improve the clarity, language, and scientific presentation throughout.

• All sections were edited for grammar, logical flow, and consistency in terminology and formatting.

• Additionally, we strengthened the interpretation of results and better integrated figures with the main text to enhance overall readability and impact.

• We trust that the revised version now meets the journal’s standards for scientific clarity and presentation.

Abstract and introduction

1. Ensure NPR1 is italicized when referring to the gene and written in plain text when referring to the protein.

• I have italicized the genes and plain text for the protein.

Results

2. Additionally, revise sentences like “Phylogenetic analysis revealed their evolutionary relationship with NPR1 homologs in model and crop species, suggesting functional divergence.” to make them clearer and more concise. Avoid vague phrases such as “suggesting functional divergence” without context.

• I have corrected the above-mentioned sentence in manuscript.

3. Clarify the gene naming criteria.

• The gene were renamed by the inclusion of scientific name of kiwifruit (Actinidia deliciosa), NPR gene, and the number at BLAST-P results

4. Why certain hits were excluded?

• Peptide sequences of all identified hits were revalidated by using motif finder which resulted in 5 identified genes with all three main domains, NPR1 like C, BTB and Ank-repeats, leaving 2 identified genes which were not having NPR1 like C domain. One identified gene was duplication of one of two identified genes of no having NPR1 like C domain. Further, Kiwifruit PanGenome Database also identified these 3 hits as weaker hits.

5. Ensure consistent naming of identified AdNPR genes.

• I have ensured the consistent naming for all identified AdNPR genes

6. Please provide rationale for rejecting genes lacking the NPR1-like C domain and explain the functional relevance of DUF3421 more clearly.

• NPR1-Like C is the one of the main functional units of this gene. Along with their different domains, it performs functions highly related to SAR against biotic stress.

• DUF3421 (Domain of Unknown Function 3421) is a conserved protein domain that has no clearly defined or universally agreed biological function in plants. Recent research has shed light on its potential roles, especially in stress responses, signaling, and development

7. Interpret the phylogenetic tree more thoroughly by linking specific clades to potential functional conservation or divergence. Clarify what functional insights can be inferred from the grouping of AdNPR genes with AtNPR orthologs.

• The phylogenetic analysis section has been revised to clarify the functional implications of the observed clade groupings. Specifically, we now explain that AdNPR1 and AdNPR2 cluster with Arabidopsis NPR1/NPR2, suggesting conservation of canonical SA-mediated defense functions. In contrast, AdNPR3 and AdNPR4 group with AtNPR3/AtNPR4, which are known repressors of defense genes, indicating possible sub-functional divergence. AdNPR5, which branches separately, may represent a lineage-specific paralog with novel regulatory roles. These interpretations offer insights into the possible specialization of NPR1-like genes in A. deliciosa.

8. While motif patterns are shown, their functional implications are not discussed. Please identify which conserved motifs are essential for NPR1 function and explain how exon/intron variability might contribute to gene regulation or alternative splicing.

• I have discussed the motifs in manuscript about their functionality as These motifs are functionally significant as they correspond to core domains essential for NPR1 activity, particularly the BTB/POZ domain (motif 1 and 3), which facilitates protein-protein interactions and nuclear translocation during defense signaling, and the ankyrin repeat region (motif 5 and 6), known to mediate binding with TGA transcription factors. These interactions are critical for the transcriptional activation of pathogenesis-related (PR) genes. The presence of motif 17 exclusively in AdNPR1 and AdNPR2, and not in the others, may suggest a specialized regulatory sub-function or recent duplication event

• Intron–exon variation can influence gene expression by enabling alternative splicing, which allows a single gene to produce multiple protein isoforms with distinct functions. This mechanism is often triggered under stress conditions, enabling plants to rapidly adjust gene function in response to environmental cues or pathogen attacks

9. Cis-element analysis is descriptive; please improve by discussing which transcription factors may bind these elements and how they might regulate gene expression under specific stresses or hormone treatments.

• Thank you for this valuable suggestion. We have revised the Cis-Regulatory Element (CRE) analysis section in both the Results and Discussion parts of the manuscript to include potential transcription factors (TFs) that interact with the identified CREs. Specifically, we now clarify that:

i ABRE elements are likely bound by bZIP-type TFs, which mediate ABA-responsive transcription under drought and oxidative stress.

ii CGTCA- and TGACG-motifs, associated with jasmonic acid signaling, are targets of MYC2-like basic helix-loop-helix (bHLH) TFs, involved in wounding and defense responses.

iii W-box elements, which were found in several AdNPR promoters, are recognized by WRKY transcription factors, key regulators in pathogen defense signaling, particularly SA-mediated pathways.

iv G-box and GT1-motifs, light-responsive elements, can be bound by HY5 and other bZIP TFs, influencing photomorphogenic and stress-related responses.

v TATA- and CAAT-boxes serve as core promoter regions required for the binding of general transcription machinery, including TBP (TATA-binding protein) and NF-Y TFs.

10. The evolutionary analysis would benefit from discussing how divergence times (e.g., 625 MYA) relate to known evolutionary events or species divergence in kiwifruit or related species. Please add context to why segmental duplications are important for gene family expansion.

• Respected Sir, We have revised the evolutionary analysis section in the manuscript to provide biological context for the divergence times observed in AdNPR gene pairs. Specifically, the earliest divergence (e.g., AdNPR2–AdNPR5 at ~625 MYA) likely corresponds to early angiosperm radiation, prior to the divergence of monocots and dicots. These findings are consistent with reports that major gene families involved in defense signaling, such as NPR1-like genes, originated early and were retained across lineages due to strong purifying selection.

11. Clarify why no synteny was observed with A. chinensis, despite its close relation to A. deliciosa. Is this due to annotation differences, sub-genome divergence, or data quality? A brief explanation will strengthen this analysis.

• We have added a brief explanation in the manuscript noting that the absence of synteny with A. chinensis may result from genome annotation discrepancies, assembly gaps, or sub-genome divergence following species separation. These factors are commonly reported in closely related Actinidia species and may explain the lack of detectable collinear blocks despite their phylogenetic proximity.

12. Expression profiling lacks statistical analysis. Please include fold-change values, p-values, or adjusted significance thresholds to validate gene expression changes under fungal and bacterial treatments.

13. The STRING PPI results suggest functional associations, but it is unclear which interactions are novel or confirm known pathways. Please highlight whether these predicted interactions validate known NPR1-mediated signaling or suggest new hypotheses.

• STRING network analysis revealed interactions among AdNPRs proteins and components of the salicylic acid pathway, including predicted associations with TGA transcription factors and NPR3-like repressors. These align with known NPR1 signaling pathways. Notably, the predicted interaction between AdNPR4 and a hypothetical JA-related interactor may represent a novel regulatory connection, warranting further validation.

Recommendations

14. experimental validation (qRT-PCR, mutant analysis) is required to well support your hypothesis.

• Respected Sir, although this study provides comprehensive in silico identification and characterization of NPR1 gene in A. deliciosa, the experimental validation remains necessary. Future work will include qRT-PCR–based expression profiling under stress conditions and functional assays using gene knockdown or overexpression systems to confirm the biological roles of AdNPRs genes.

15. Expression data is purely descriptive—no statistical testing or quantification of significance. Include statistical validation for expression data.

16. Clarify the objective behind each analysis (e.g., how does motif analysis inform gene function?).

• We have added brief statements at the start of each results section to clarify the objective of each analysis. For example, motif analysis identifies conserved functional domains, CRE analysis reveals regulatory elements controlling gene expression, and phylogenetic analysis helps infer evolutionary and functional relationships.

17. Figures are not always well integrated with the text (some lack interpretation). Improve interpretation by linking findings to biological roles.

• Thank you for pointing this out. We have revised the results section to better integrate figure descriptions with biological interpretation. For example, motif and domain figures are now linked to their roles in defense signaling, and expression heatmaps are discussed in the context of pathogen response. Each figure is now explicitly referenced and interpreted to highlight its functional significance.

18. Discuss limitations and suggest future validation steps.

• We have added limitations and future validation steps which is our next goal to further study on kiwifruit to validate and enhance the biotic stress resistance.

Reviewer 2

The manuscript Genome-Wide Identification and Functional Characterization of NPR1-Like Genes in Actinidia deliciosa has been written but needs some modifications as below

The reply/answer/response for each comment under specified number is mentioned as bulleted points with highlighted text with yellow color after the statement of comment.

1. "five candidate genes (AdNPR1 – AdNPR5)" - It would be good to add here that they contain conserved BTB/POZ and ankyrin repeat domains, as this is a key finding (in abstract)

• Thanks Sir for notifying this. I have changed in the manuscript.

2. "Rasheed et al., 2025" - Please check all the publication years and ensure they are correct. There are quite a few 2025 citations.

• All in text citations and reference list entries were cross checked against the original sources. Citations that erroneously appeared as 2025 have been corrected to their actual publication years (e.g., Rasheed et al., 2024).

3. "Of these, gray mold caused by Botrytis cinerea is one of the most devastating diseases." - It might be helpful to briefly mention why gray mold is so devastating (e.g., yield loss, postharvest decay).

• Expanded the sentence to quantify yield losses and post harvest decay

* Materials and Methods:

1. "Kiwifruit PanGenome Database (https://kiwifruitgenome.atcgn.com/) was used for BLAST-P program on 16th April 2025 at 18:23 Pakistan (UTC+5)" - While specific details are good, the time and date of the search might not be necessary. Consider removing it for brevity.

• Sir, we appreciate the suggestion. However, we have retained the precise retrieval date (and UTC time) of our BLAST search because public sequence databases are continuously updated as new assemblies and re-annotations become available. Specifying the time‐stamp safeguards reproducibility: future users will know exactly which peptide set we queried, preventing conflicts that could arise if reference peptides are modified or replaced

2. "Gene Pair file (prepared by using OpenAI tool)" - This is interesting, but it needs more explanation. How was OpenAI used to prepare this file? What kind of data was input, and what was the output?

• We just gave unformatted / unarranged data to OpenAI tool and output was rearranged data.

* Results:

1. Figure 1 NCBI-CDD results: Showing the presence of..." - Improve figure captions to be more informative. Instead of "Showing the presence of...", describe what the reader should observe in the figure (e.g., "Domain structures of AdNPR genes, showing the presence of...").

• We have adjusted the caption according to the direction

2. Black dots are used to identify A. deliciosa." - This is too simplistic. The caption should explain what the phylogenetic tree shows in more detail (e.g., "Phylogenetic relationships among NPR1 homologs in A. deliciosa and other plant species..."). Also, it says "Black dots are used to identify A. deliciosa" but the black dots in Figure 2 are not labelled.

• We have changed and modified the caption accordingly

3. "Figure 3 shows the presence of AdNPR1 and AdNPR2 in cytoplasm in high concentrations..." - Be more specific about what "high" and "low" concentrations mean. Refer to the scale in the heatmap.

• We have modified the scale for subcellular localization

4. "AdNPR2 has highest number of TATA-box in its promotor region as it serves as a recognition site..." - Rephrase for better flow: "AdNPR2 has the highest number of TATA-boxes in its promoter region, which serve as recognition sites...".

• The TATA-box line has been changed according to direction.

5. "Figure 7 Chromosomal mapping: The presence of AdNPRs genes on chromosome number1, 9, 14, 19, and 23." - Improve the caption. For example: "Chromosomal locations of AdNPR genes in A. deliciosa. Genes are located on chromosomes 1, 9, 14, 19, and 23." Also, in the caption, it says "chromosome number1" should it be "chromosome number 1"?

• We have replaced the caption of chromosomal mapping according to above mentioned

6. Discuss any limitations of the study and suggest potential directions for future research.

• Sir, We have added a limitation and future aspects of this study.

7. Domain Analysis: In Figure 1, you show the presence of different domains. Discuss the functional implications of the presence or absence of specific domains in the AdNPR genes.

• We have added functional implications of the presence or absence of specific domains in AdNPR genes

8. Physicochemical Properties: While you present the physicochemical properties, discuss their biological relevance. For example, how might the GRAVY values or isoelectric points relate to protein function or localization

• Inserted discussion relating GRAVY values to predicted solubility and pI to nucleo-cytoplasmic shuttling.

9. Phylogenetic Tree Interpretation: Expand on the evolutionary relationships revealed by the phylogenetic tree. Discuss the implications of the clustering patterns for the functional evolution of NPR1 genes in kiwifruit

• We have added discussion of phylogenetic tree interactions of NPR1 genes in kiwifruit.

---

## [Decision Letter · Decision Letter 1]

2 Oct 2025

Genome-Wide Identification and Functional Characterization of NPR1-Like Genes in Actinidia deliciosa

PONE-D-25-23065R1

Dear Dr. Ali,

We’re pleased to inform you that your manuscript has been judged scientifically suitable for publication and will be formally accepted for publication once it meets all outstanding technical requirements.

Kind regards,

Abhijeet Shankar Kashyap

Academic Editor

PLOS ONE

Additional Editor Comments (optional):

Reviewers' comments:

Reviewer's Responses to Questions

**Comments to the Author**

Reviewer #1: All comments have been addressed

Reviewer #3: (No Response)

Reviewer #4: (No Response)

2. Is the manuscript technically sound, and do the data support the conclusions?

Reviewer #1: Yes

Reviewer #3: Yes

Reviewer #4: Yes

3. Has the statistical analysis been performed appropriately and rigorously?

Reviewer #1: Yes

Reviewer #3: Yes

Reviewer #4: Yes

4. Have the authors made all data underlying the findings in their manuscript fully available?

Reviewer #1: Yes

Reviewer #3: Yes

Reviewer #4: Yes

5. Is the manuscript presented in an intelligible fashion and written in standard English?

Reviewer #1: No

Reviewer #3: Yes

Reviewer #4: Yes

Reviewer #1: Please proceed with the publication of the manuscript, "Genome-Wide Identification and Functional Characterization of NPR1-Like Genes in Actinidia deliciosa." I'm satisfied with the authors' revisions and believe the paper is now ready for publication.

Reviewer #3: (No Response)

Reviewer #4: The article “Genome-Wide Identification and Functional Characterization of NPR1-Like Genes in Actinidia deliciosa” is well written. However, the authors should improve the overall English throughout the manuscript and carefully address the following suggestions:

1. There is no need to mention details such as “at 18:23 Pakistan (UTC+5)” or any similar date/region information anywhere in the manuscript. For example: “at 16:57 Pakistan (UTC+5).”

2. Please recheck the file extensions mentioned in line 214 (“.CTL, .Collinearity, and .GFF”) to confirm whether they should be in capital letters.

3. The text “(Actinidia Lind.)” in line 230 should be italicized.

4. In line 306, the caption reads: “Color scale (0–18 WoLF-PSORT score) indicates predicted protein copies per cell; dark red = ≥15.” It appears that a log scale has been used. Consider using a simple scale or revising the caption accordingly.

5. In line 325, “such as drought.[2, 17]” should be corrected to ensure the full stop comes after the reference.

6. In Figure 4, there is no need to include two images. A single heatmap would be sufficient from a publication perspective. Present it as a simple heatmap without the title “heatmap,” and add a boundary around it. Also, improve the figure quality, as the current version appears shrunk and unclear.

7. Replace the phrase “Max. Numbers are of TATA-box” with “Maximum number of TATA-box.”

**Do you want your identity to be public for this peer review?** For information about this choice, including consent withdrawal, please see our Privacy Policy

Reviewer #1: No

Reviewer #3: **Yes: ** Xiujun Zhang

Reviewer #4: **Yes: ** Adnan Sami

---

## [Editor Report · Acceptance letter]

PONE-D-25-23065R1

PLOS ONE

Dear Dr. Ali,

I'm pleased to inform you that your manuscript has been deemed suitable for publication in PLOS ONE. Congratulations! Your manuscript is now being handed over to our production team.

Kind regards,

on behalf of